# In Situ Study of the Magnetic Field Gradient Produced by a Miniature Bi-Planar Coil for Chip-Scale Atomic Devices

**DOI:** 10.3390/mi14111985

**Published:** 2023-10-26

**Authors:** Yao Chen, Jiyang Wang, Ning Zhang, Jing Wang, Yintao Ma, Mingzhi Yu, Yanbin Wang, Libo Zhao, Zhuangde Jiang

**Affiliations:** 1State Key Laboratory for Manufacturing Systems Engineering, International Joint Laboratory for Micro/Nano Manufacturing and Measurement Technologies, Overseas Expertise Introduction Center for Micro/Nano Manufacturing and Nano Measurement Technologies Discipline Innovation, School of Mechanical Engineering, Xi’an Jiaotong University, Xi’an 710049, China; yaochen@xjtu.edu.cn (Y.C.); jiyangwang@stu.xjtu.edu.cn (J.W.); myt1064212021@stu.xjtu.edu.cn (Y.M.); yumingzhi@stu.xjtu.edu.cn (M.Y.); wybaai@stu.xjtu.edu.cn (Y.W.); libozhao@xjtu.edu.cn (L.Z.); zdjiang@xjtu.edu.cn (Z.J.); 2Xi’an Jiaotong University Suzhou Institute, Suzhou 215123, China; 3Research Center for Quantum Sensing, Intelligent Perception Research Institute, Zhejiang Lab, Hangzhou 311100, China; 4Beijing Institute of Electronic System Engineering, Beijing 100854, China; wangjing_vicky@buaa.edu.cn

**Keywords:** miniature bi-planar coil, magnetic field gradient, atomic magnetometer, chip-scale quantum sensors, optical pumped magnetometer

## Abstract

The miniaturization of quantum sensors is a popular trend for the development of quantum technology. One of the key components of these sensors is a coil which is used for spin modulation and manipulation. The bi-planar coils have the advantage of producing three-dimensional magnetic fields with only two planes of current confinement, whereas the traditional Helmholtz coils require three-dimensional current distribution. Thus, the bi-planar coils are compatible with the current micro-fabrication process and are quite suitable for the compact design of the chip-scale atomic devices that require stable or modulated magnetic fields. This paper presents a design of a miniature bi-planar coil. Both the magnetic fields produced by the coils and their inhomogeneities were designed theoretically. The magnetic field gradient is a crucial parameter for the coils, especially for generating magnetic fields in very small areas. We used a NMR (Nuclear Magnetic Resonance) method based on the relaxation of ${}^{131}Xe$ nuclear spins to measure the magnetic field gradient in situ. This is the first time that the field inhomogeneities of the field of such small bi-planar coils have been measured. Our results indicate that the designed gradient caused error is 0.08 for the $B_{y}$ and the $B_{x}$ coils, and the measured gradient caused error using the nuclear spin relaxation method is 0.09±0.02, suggesting that our method is suitable for measuring gradients. Due to the poor sensitivity of our magnetometer under a large $B_{z}$ bias field, we could not measure the $B_{z}$ magnetic field gradient. Our method also helps to improve the gradients of the miniature bi-planar coil design, which is critical for chip-scale atomic devices.

## 1. Introduction

Miniaturization of atomic devices [1], such as atomic magnetometers [2,3,4,5,6,7,8,9,10], atomic gyroscopes [11,12,13], atomic clocks [14,15], micro-fluidic channels [16,17,18], etc., promotes the use of quantum technology sensors in wide-ranging applications, such as magnetoencephalography (MEG), magnetocardiography(MCG), navigation, time keeping, and biomedical sensing. These atomic devices are mainly composed of an alkali vapor cell, optical components, lasers, photodetectors, and magnetic field coils. The magnetic field is typically used for spin modulation and field compensation [2], holding magnetic field generation [12,15], and spin manipulation [17]. The magnetic field coils are usually mounted on the outermost part of the sensor head and can account for a large part of the sensor volume. It is difficult to miniaturize the typical three-axis coils, such as the Helmholtz coil. However, the bi-planar coil, which can produce three-dimensional magnetic fields with only two parallel planes of current confinement [19], gives us the opportunity to further miniaturize the devices. As we know, it is hard to fabricate a 3D Helmholtz coil in the micro-machined atomic magnetometer. The typical micro-machining process includes etching, bonding and film deposition, which are not compatible with the 3D coil fabrication. However, the bi-planar coil fabrication is suitable for the micro-machined process. The bi-planar coil may be widely used in chip-scale atomic devices. It was reported that the miniature bi-planar coil has been used in atomic magnetometers [20]. After years of studying atomic magnetometer and gyroscope miniaturization, we believe that the bi-planar coils are suitable for a compact design of atomic devices.

The bi-planar coil, which is typically used to produce stable magnetic fields or magnetic field gradients for field cancellation [9], has been widely used in Magnetic Resonance Imaging systems [19,21]. With the development of optically pumped magnetometers for MEG or MCG recording, the bi-planar coil was designed for field compensation in a magnetic field shield room [22,23]. In order to measure the brain’s magnetic field as a person’s head is moving, a 1.6m×1.6m bi-planar coil system was used to compensate for the moving-related background magnetic field fluctuation. The coil design and fluctuation field compensation were thoroughly studied [22,24]. The high uniform compensation magnetic field requirements in the brain magnetic field measurement area have driven researchers to optimize the bi-planar design [25,26,27,28]. A specially designed algorithm was even presented in a reference [27].

The magnetic field gradient is a critical parameter for atomic devices, as it can cause spin relaxation, leading to broadened response lines that can worsen sensitivity. Additionally, gradient relaxation can reduce the polarization of hyper-polarized nuclear spins. For MEG measurement, a bi-planar coil with gradient compensation coil was designed for better uniformity. It is easy to map the magnetic field gradient in a large bi-planar coil mounted in a magnetic field shielding room [29], and the bi-planar coil was also designed with an optimization algorithm to improve uniformity [27]. However, for a bi-planar coil used for miniature devices, it is impossible to measure the gradient using traditional methods. Furthermore, even if a small bi-planar coil is fabricated based on a larger optimized coil, factors such as the line width of the coil wire can degrade the uniformity of the coil magnetic field. Therefore, it is important to measure the uniformity of the coil in situ.

As we know, the relaxation of the hyperpolarized nuclear spin such as ${}^{131}Xe$ is closely related to the magnetic field gradient. This gives us the chance to solve the problem in our study. In this paper, we can use the relaxation of the nuclear spin for measurement of the bi-planar coil magnetic field gradient. There have been several comprehensive studies on the nuclear spin relaxations induced by magnetic field gradients [30,31]. The relaxation is closely related to the size of the vapor cell and the buffer gas pressure in the vapor cell, which determine the time of nuclear spin movement across the vapor cell relative to the precession time of the nuclear spin in one period. In this paper, we used the relaxation method to measure the magnetic field gradient produced by the bi-planar coil. Based on the theories shown in these references [30,31,32,33,34], we varied the magnetic field gradients, and then the relaxations of Xe atoms were measured. The measured magnetic field inhomogeneity was compared with the simulation results of the bi-planar coils. The results show that we can successfully measure the inhomogeneity of the coil. This is the first time that the magnetic field gradient related relaxation of the nuclear spins is used for the miniature coil magnetic field gradient measurement. Our study aims to contribute to the fabrication of bi-planar coils with better uniformity for miniature atomic devices.

## 2. Theory

### 2.1. Coil Design

We aim to produce a constant magnetic field in the *x*, *y*, and *z* directions. A bi-planar coil can produce a three-dimensional magnetic field and confine the current of the coil to just two planes. There are three sets of double-plane coils, two in each set, facing each other to generate a magnetic field in one direction at their central position. Similarly, the other two sets of coils are placed in the same way, and all three sets of coils are located in two opposite planes. This creates a three-dimensional magnetic field at the center. Therefore, compared with the three-dimensional structure of the Helmholtz coil, it can greatly reduce the size of the sensor head.

In our experiment, the vapor cell is a cylinder with an inner diameter of 5 mm and a length of 4 mm. Therefore, we need to produce a uniform magnetic field in this small area that is as flat as possible. A bi-planar coil needs to be confined to two planes, so it is convenient to choose the center point between the two planes in which the vapor cell should be placed.

To design the coil, we first need to select a target region, similar to the region in this reference [27]. We chose the target area to be 4mm×4mm×4mm, which is approximately the same as the volume of the vapor cell. The length of the bi-planar coil was designed to be 16 mm. The basic idea behind the bi-planar coil design is to first determine the desired magnetic field in the target area. In this study, a uniform magnetic field is desired. The current distribution on the plane is obtained through several steps of inverse calculation. We will not go into detail about this calculation process but rather provide the results of the coil design.

The bi-planar coil is traditionally used for Magnetic Resonance Imaging (MRI) coil design, and details about the coil design can be found in references [19,21,35]. The bi-planar coil is also used for residual magnetic field compensation in a magnetic field shielding room, and details on how to design such a coil are demonstrated in references [22,24,26,36].

Figure 1 and Figure 2 display the design of the bi-planar coils for $B_{x}$, $B_{y}$, and $B_{z}$. The red and blue lines represent different directions of current flow. The $B_{x}$ and $B_{y}$ coils share the same design drawing. The magnetic field strengths in the target regions are 10 nT, and the calculated coil constants for the coils are 42 nT/mA, 42 nT/mA, and 476 nT/mA, respectively.

### 2.2. Theory of the Gradient Measurement: Relaxation of Nuclear Spins under Inhomogeneity Magnetic Field

It is well known that the inhomogeneity of the magnetic field can cause the relaxation of hyper-polarized nuclear spins. Both the longitudinal relaxation rate $1/T_{1}$ and the transverse relaxation rate $1/T_{2}$ may be affected by the magnetic field gradients. The magnetic field inhomogeneity-induced relaxation is related to the buffer gas pressure of the vapor cell. Typically, two conditions are considered as we study the relaxation, namely high buffer gas pressure and low buffer gas pressure [32]. This pressure could affect the diffusion time of nuclear spins in the vapor cell. Under low pressures, the nuclear spins could move quickly in the vapor cell, and there is a characteristic diffusion time $\tau_{d}$ to account for this process. In a spherical vapor cell with a radius of *R*, this time is approximately $R^{2}/4D$ [37]. *D* is the diffusion coefficient of nuclear spins in the buffer gas. Under the low-pressure limit, the diffusion time of nuclear spins should be much smaller than the nuclear spin precession period. The nuclear spins should experience many times of moving to the vapor cell wall in one precession period. We define the precession time $\tau_{l}$ to be $2\pi /\gamma B_{0}$. $B_{0}$ is the average magnetic field in the volume of the vapor cell. Under the low-pressure limit, $\tau_{d}$ should be much smaller than $\tau_{l}$. Thus, it is easy to understand the high-pressure limit in which $\tau_{d}$ should be much larger than $\tau_{l}$. We can find excellent explanations and the theory of this process in reference [32].

In this study, our main focus is on the transverse relaxation of nuclear spins. The holding magnetic field, under which we measure the transverse relaxation of the ${}^{131}$Xe nuclear spin, is around 500 nT. Based on the nuclear spin’s gyro-magnetic ratio γ, the precession period in our experiment is around 0.5 s. The vapor cell is filled with 85 KPa of nitrogen gas. Using the diffusion coefficient of Xe in $N_{2}$ under 760 Torr $N_{2}$ at 455 K ($D_{0}\left(Xe-N_{2}\right)=0.21\;{\mathrm{cm}}^{2}/s$ [31]), we estimate the diffusion time to be around 0.08 s. Note that the diffusion constant is weakly temperature dependent. The measuring temperature for the gradients is around 393 K. The diffusion constant could be considered to be similar to that of the 455 K condition. Therefore, our experiment is under the low-pressure limit.

There are several references that discuss the theory of the magnetic field gradient-induced relaxation of nuclear spins [31,32,33,37]. Some studies that focus on experimental investigations can also be found in references [30,34,38]. We will not repeat the detailed theory here. Under low pressures, the transverse relaxation rate can be expressed as:(1)$$\frac{1}{T_{2}}=\frac{4R^{4}}{175D}\left(|\vec{\nabla }\Omega_{Ix}{|}^{2}+|\vec{\nabla }\Omega_{Iy}{|}^{2}+2|\vec{\nabla }\Omega_{Iz}{|}^{2}\right)$$$\Omega_{I}$ represents the angular precession frequency of the nuclear spins under the influence of the inhomogeneous magnetic field. We define the average magnetic field across the entire vapor cell as $B_{0}$, which is aligned in the *z* direction. The actual magnetic field $\vec{B}\left(\vec{r}\right)$ is position-dependent, and the inhomogeneous magnetic field is given by $\vec{B_{I}}=\vec{B}\left(\vec{r}\right)-B_{0}\vec{z}$. The Larmor precession induced by the inhomogeneous magnetic field is given by $\vec{\Omega_{I}}=\gamma \vec{B_{I}}$. The gradient of $\Omega_{Ix}$ with respect to *x* is denoted by $\vec{\nabla }\Omega_{Ix}=\left(\partial \Omega_{Ix}/\partial x,\partial \Omega_{Ix}/\partial y,\partial \Omega_{Ix}/\partial z\right)$. Note that in our experiment, the vapor cell is cylindrical with a diameter that is approximately equal to its length, so it is reasonable to assume that the vapor cell resembles a sphere.

## 3. Experimental Setup

### 3.1. The Measurement Platform

The homemade vapor cell used in this paper is cylindrical in shape. A small amount of Cs metal is filled in the vapor cell. Additionally, we filled the vapor cell with 750 Pa ${}^{131}Xe$ for measuring the bi-planar coil parameters. We utilized a 3 W, 1550 nm heating laser for heating the vapor cell to the desired temperature as the optical depth was approximately 1.5. The heating laser was transferred through a 400 μm core fiber. We used PEEK thermal insulation, and the structure was manufactured through 3D printing.

The 894 nm pumping laser was coupled to a polarization-maintaining fiber through a fiber coupling port. The pumping laser light was collimated by lens1. The λ/4 waveplate after lens1 changed the linearly polarized light into circularly polarized light. PD2 received the pumping laser light, which passed through the vapor cell. The pumping laser light polarization was purified by the GT prism before going to the fiber coupling port. The combination of the λ/2 waveplate and PBS was able to sample a small part of the pumping laser, which was detected by PD1. To balance the optical power hitting the detectors, we used an NDF to adjust the power of the sampling laser. We used the sampling laser to balance the background offset of the pumping laser after the vapor cell. Specifically, the sampling laser was detected before the fiber by PBS and λ/2, and the light detected on PD1 was subtracted from PD2 to eliminate background offset. To eliminate the polarization drifts, we need to align the polarization direction of the pump laser with the direction of the slow axis or fast axis of the PM.

We used a homemade differential circuit to subtract the background offset in PD2. Several layers of magnetic field shields were used to shield the outside magnetic field. A cosθ coil with determined coil constants was used to calibrate the bi-planar coil. The bi-planar coils were wrapped around the vapor cell with a distance of 15 mm between them. The three sets of bi-planar coils were able to produce magnetic fields in three directions. The pumping laser direction was along the *y* axis. We did not need to move the cell for each position as we measured the coil constants or the gradients. During the measurement, the vapor cell was always located at the center of the coil and was 7.5 mm away from the coil on both sides. This is why we think that our method can be in situ. Moreover, it is impossible to move a tiny magnetometer in such a small area.

### 3.2. Coil Fabrication

We designed and fabricated a flexible PCB for the bi-planar coil. Since the coils in the Matlab R2020a are independent rings, we needed to use several wires to connect all the rings in series to form a pathway in the actual processing. Additionally, PCB wiring cannot be crossed, so we needed to design a multi-layer board structure. This means that there is a difference between the actual wiring diagram and the theoretical one. We completed the PCB wiring with a two-layer board structure. Each coil had four pairs of current inlet and outlet, and during installation, only the respective four pairs of ports needed to be connected in series. The two coils in each direction were powered by a signal generator, and a total of three signal generators were able to make the bi-planar coil work normally. It is worth noting that the wire width is only 2 mil, so the side length of the coil is only 16 mm, which is much smaller than that of traditional Lee-Whiting and cosθ coils. We used a flexible circuit board (FPC) to fabricate the bi-planar coil because its advantages of being small in size and lightweight make it more suitable for miniaturization.

Figure 3 shows the PCB drawing of the bi-planar coil. A uniform magnetic field is designed to be generated at the center of the coil. What is more, there are lines which did not form the theoretical part of the coil and were needed for the current connection. We tried to make the current in these lines parallel and opposite so that the disturbing magnetic fields generated by them can cancel each other. Moreover, there are several soldering ports in the coil design which could also contribute to field distortion. These ports are designed to flow currents in the opposite direction in the adjacent ports. Note that these ports are also far away from the main body of the coils. Therefore, we believe that the influence of the currents generated by the inlet and outlet ports can be ignored.

### 3.3. Sensor Head Fabrication

The sensor head is used to hold the vapor cell, various optical elements, and the optical fiber in place. Therefore, the structure and size of the optical path must be taken into consideration. Due to the complexity of the sensor head’s structure, we used 3D printing to fabricate it. The internal optical path of the sensor head is shown in the right part of Figure 4, where two right-angle prisms are placed in the optical path to fold it and reduce the volume of the sensor head further. Please note that the vapor cell is located in the center of the coil and is 7.5 mm away from the coil on both sides. So, our method of measuring the coils’ gradient is in situ. The size of the sensor head is only 47mm×32mm×18mm.

PEEK material was chosen to fabricate the sensor head because it is a special engineering plastic with high temperature resistance and high mechanical strength, making it suitable for processing machine parts. The vapor cell’s working temperature is about 120 °C, so the PEEK material meets the requirements.

## 4. Results

### 4.1. Coil Constants Measurement

The coil constant is defined as the magnetic field produced by the coil at a certain current inside the coil. We need to measure the magnetic field as we change the coil current. We know that if there is a holding magnetic field in the optical pumping direction *y* and a pulse signal drives the nuclear spins to deviate from the pumping direction, the nuclear spins will precess around the holding magnetic field $B_{y}$. The precession frequency is linearly related to the holding magnetic field, and we can use this method to measure the coil constant.

As shown in Figure 5, we recorded the Free Induction Decay (FID) signal at −10 mA of the coil, which is typically used for relaxation measurement and even for the measurement magnetic field [3]. The FID signal is caused by the precession of the ${}^{131}$Xe nuclear spins around the holding magnetic field. We fitted the signal to a sinusoidal function multiplied by an exponential decay function [41,42], and we calculated the precession frequency and the decay rate of the FID signal to be 1.4 Hz and 0.50$\;s^{-1}$, respectively. According to the gyro-magnetic ratio of ${}^{131}$Xe, which is 0.0042 Hz/nT, the total magnetic field can be calculated to be 381 nT.

Then, we recorded the precession frequency of the nuclear spins at different coil currents. Finally, we obtained the total magnetic field experienced by the nuclear spins. Figure 6 shows the relationship between the current and the magnetic field. We can deduce that the coil constant of the bi-planar coil in the *y* direction is 51 nT/mA. This value is approximately equal to the simulated coil constant, which is 42 nT/mA. Note that the configuration of the coil in the *x* and *y* directions are the same. The coil constant in the *z* direction was also measured by the nuclear spin precession method. Please see the result in Table 1.

For further comparison, we used cosθ coils in the *z* and *y* axes and a Lee-Whiting coil in the *x* axis, whose coil constants are known and calibrated, to calibrate the bi-planar coil as shown in Figure 4. The coil constants of the cosθ coils and the Lee-Whiting coil were found to be 80 nT/mA and 240 nT/mA, respectively.

The experimental setup can also function as an atomic magnetometer, producing output voltage signals when a magnetic field is inputted. The magnetometer is similar to the one described in this reference [2]. To calibrate the bi-planar coil, we varied the input currents of the Lee-Whiting coil, which served as the calibration coil for the *x* direction, and recorded the output voltages, resulting in a linear relationship between input current and output voltage with a slope of 5979 mV/mA. To calibrate the *x* direction bi-planar coil, we repeated this process, and the slope was 1226 mV/mA. At the same input current, the output voltage of the calibration coil was 4.9 times that of the bi-planar coil. Based on the coil constant of the Lee-Whiting coil, we deduced that the constant of the bi-planar coil was 49 nT/mA, which is similar to the measured coil constant of 51 nT/mA using the precession method. Note that the *x* and *y* direction bi-planar coils should have the same coil constant. The theoretical value of the coil constant was 42.4 nT/mA, which is approximately equal to our measured coil constant.

As for the bi-planar coil in the *z* direction, we used the same method. The difference is that the calibration coil used in the *z* direction is of the cosθ type. The result is shown in Table 2.

We also measured the FID signal of ${}^{131}$Xe nuclear spins using a larger coil system with more uniform magnetic field gradients. Due to the smaller gradients of the coil, the relaxation time of the nuclear spins was longer, allowing us to observe the resolved nuclear quadrupolar-induced splitting of the nuclear energy levels. The beating signal was clearly visible. Figure 7 shows the FID signal obtained. We fitted the signal and determined that the relaxation rate was 0.32 $s^{-1}$. We also calculated the quadrupolar frequency shift to be 0.17 Hz and the precession frequency of the nuclear spin to be 2.0 Hz.

We also measured the relaxation of ${}^{131}$Xe under a magnetic field in the *z* direction. This allowed us to measure the gradient of the bi-planar coil through relaxation. We first set a holding magnetic field in the *y* direction and compensated for the magnetic fields in the other two directions. Then, the nuclear spins could be polarized in the *y* direction. After about two minutes of hyper-polarization, we suddenly changed the current in the *z* direction to 10 mA and withdrew the holding magnetic field in the *y* direction simultaneously. The *y*-stretched nuclear spins would precess around the *z* direction magnetic field produced by the large cos θ coil.

As shown in Figure 8, we fitted the experimental data to a combination of several sinusoidal signals. The measured relaxation rate of ${}^{131}$Xe was 0.30 $s^{-1}$. This rate was similar to the measured rate as the nuclear spins processed around the *y* direction magnetic field as shown in Figure 9. We measured the relaxation rate of ${}^{131}$Xe to be 0.50 $s^{-1}$ in the presence of the magnetic field from the *z* direction bi-planar coil. The result is shown in Table 3. There was an obvious shortening of the relaxation time due to the gradient of the bi-planar *z* coil.

Moreover, we measured the coil constant of the *z* bi-planar coil through the precession of ${}^{131}$Xe nuclear spins. As we illustrated, we measured the precession frequency under different currents, which were injected into the *z* direction bi-planar coil. The measured result was 574 nT/mA. This result was approximately the same as the 535 nT/mA, which was measured by the calibration coil method earlier.

### 4.2. Magnetic Field Gradient-Induced Relaxation Measurement

The coil is designed assuming a uniform magnetic field in the target area, but in reality, there is a magnetic field gradient in the target area. As we mentioned in the ’coil design’ subsection, we can inversely acquire the current distribution in the plane of the coil based on the magnetic field in the target area. After the current has been acquired, we can calculate the real magnetic field in the target area numerically based on the Maxwell equations. Note that the target area is assumed to be a constant magnetic field. The actual magnetic field is actually changing with the positions. The design of the biplanar coil actually makes some approximations during the design process. It is the Fourier transformation method which is used to decompose the target constant magnetic field into several sinusoidal functions. We also calculated the field gradient in our design. Assuming that the target magnetic field in the area is $B_{0}$, and the actual magnetic field is B(x,y,z), which is position-dependent, we define the error magnetic field at position (x,y,z) as:(2)$$\epsilon \left(x,y,z\right)=\left(B\left(x,y,z\right)-B_{0}\right)/B_{0}$$

Using this definition, we can calculate the magnetic field gradient. Figure 10 and Figure 11 show the results of the magnetic field gradient calculation in the *x*, *y*, and *z* directions.

The plots in Figure 10 and Figure 11 do not clearly show the percentage error change with position in different directions. Therefore, we provide a clearer figure depicting the error change. As illustrated in Figure 12, we set the *z* and *x* positions to 0 and then acquired the errors of the magnetic field in the *y* direction. We plotted the relationship between the *y* position and the errors, as well as the *z* and *x* errors with the change of position. It is worth noting that Figure 12 provides more details about the magnetic field for the *y* bi-planar coil. The maximum magnetic field error can reach up to 15% in the target area. We also provide a more detailed plot of the magnetic field gradient for the *z* bi-planar coil in Figure 13. The maximum error can reach up to 7% in the target area.

The magnetic field in the *z* direction exhibits near symmetry around the *z* axis. As presented in reference [33], the *z* coil produces only the magnetic field gradient components of $\partial B_{x}/\partial x$, $\partial B_{y}/\partial y$, and $\partial B_{z}/\partial z$, and only these three components contribute to the magnetic field gradient-induced relaxation. Figure 10 indicates a similar situation for the *x* direction bi-planar coil, where $\partial B_{x}/\partial x$, $\partial B_{y}/\partial y$, and $\partial B_{z}/\partial z$ also contribute to the relaxation.

According to Equation (Equation 1), the transverse relaxation of the nuclear spins could be further simplified to be:(3)$$\frac{1}{T_{2z}}=\frac{4R^{4}}{175D}\gamma^{2}\left(|\partial B_{x}/\partial x{|}^{2}+|\partial B_{y}/\partial y{|}^{2}+2|\partial B_{z}/\partial z{|}^{2}\right)$$
for the *z* direction bi-planar coil. For the *x* and *y* direction bi-planar coil, the transverse relaxation are:(4)$$\frac{1}{T_{2x}}=\frac{4R^{4}}{175D}\gamma^{2}\left(|\partial B_{z}/\partial z{|}^{2}+|\partial B_{y}/\partial y{|}^{2}+2|\partial B_{x}/\partial x{|}^{2}\right)$$
(5)$$\frac{1}{T_{2y}}=\frac{4R^{4}}{175D}\gamma^{2}\left(|\partial B_{z}/\partial z{|}^{2}+|\partial B_{x}/\partial x{|}^{2}+2|\partial B_{y}/\partial y{|}^{2}\right)$$

Based on Equation (Equation 3) and the magnetic field gradient results in Figure 11 and Figure 13, we can calculate (assuming $B_{0}$ equals to 198 nT, which is the magnetic field added to the nuclear spin from Figure 9) that the magnetic field gradient is 59 nT/cm for $\partial B_{z}/\partial z$ and 69 nT/cm for $\partial B_{x}/\partial x$ and $\partial B_{y}/\partial y$. Here, we assume a linear dependence of the magnetic field error on the position in Figure 13 for approximation. The relaxation due to the magnetic field gradient is 0.02 $s^{-1}$. Similarly, based on Equation (Equation 5) and the magnetic field gradient results in Figure 10 and Figure 12, we can calculate (assuming $B_{0}$ equals to 381 nT, which is the magnetic field added to the nuclear spin from Figure 5) that the magnetic field gradient is 285 nT/cm for $\partial B_{z}/\partial z$, $\partial B_{x}/\partial x$, and $\partial B_{y}/\partial y$. The relaxation due to the magnetic field gradient is 0.1 $s^{-1}$.

From Equation (Equation 5), we can see that the relaxation rate is dependent on the magnetic field gradient. To further study the magnetic field gradient of the bi-planar coil in the *x* or *y* direction, we can simplify Equation (Equation 5) to be:(6)$$\frac{1}{T_{2y}}=\frac{4R^{4}}{175D}\gamma^{2}*4*k_{0}^{2}B^{2}$$

Equation (Equation 6) assumes the same magnetic field gradient in all directions as shown in Equation (Equation 5). This is supported by the results presented in Figure 12. *B* represents the magnetic field strength produced by the coil, while $k_{0}$ is the coefficient that requires fitting. The factor 4 indicates that the magnetic field gradients are equal, with a total of 4. Moreover, if we change the total magnetic field added by the bi-planar coil, the magnetic field gradient could also change. That means the magnetic field gradient is dependent on the total magnetic field. Thus, we also measured the relationship between the total magnetic field and the relaxation rate of the nuclear spins. Figure 14 shows the results. We measured these results in a large magnetic field range, including both positive and negative magnetic field ranges. We fitted the measured results to Equation (Equation 5), and the dashed line in the positive range has a coefficient of $\left(2.5\pm 0.5\right)\times 10^{-7}\;s^{-1}/\;{\mathrm{nT}}^{2}$, while in the negative range, the solid fitted curve has a coefficient of $\left(5.5\pm 0.6\right)\times 10^{-7}\;s^{-1}/\;{\mathrm{nT}}^{2}$. The error in the coefficient is from the fitting process. The relaxation rates tend to be 0.31$\;s^{-1}$ and 0.31$\;s^{-1}$ as the magnetic field equals 0 for the positive and negative magnetic field ranges, respectively. This means that the residual relaxation without magnetic field gradient is equal to 0.31$\;s^{-1}$ and 0.31$\;s^{-1}$. The two numbers are approximately equal to the measured relaxation rates (0.30$\;s^{-1}$) when the big coil is utilized. This also implies that the big coil utilized in our experiment owns a very uniform magnetic field.

The radius of the vapor cell is 0.25 cm, and the diffusion coefficient for our experimental conditions is 0.18 ${\mathrm{cm}}^{2}/s$. The coefficient $4R^{4}\gamma^{2}*4/\left(175D\right)$ was calculated to be $1.3\times 10^{-6}$. Based on the fitted coefficient of $\left(2.5\pm 0.5\right)\times 10^{-7}\;s^{-1}/\;{\mathrm{nT}}^{2}$ in the positive region (according to Equation (Equation 6)), we can determine that $k_{0}$ is equal to 0.44±0.10/cm.

From Figure 12, we can see that at 0.2 cm, the error is 0.15, as well as seeing that the relationship between position and error is not linear. We define $\varepsilon_{avg}$ to be the average error of the coil. From Equation (Equation 6), the magnetic field gradient $\epsilon_{avg}B/0.2$ is equal to $k_{0}B$. Thus, $k_{0}$ includes the error of the coil and the relative distance from the center of the coil.

Finally, we can calculate that $\varepsilon_{avg}$ is 0.09±0.02. In Figure 12, the error is 0.15 at 0.2 cm. The average error of the coil should be smaller than the largest error at 0.2 cm. Therefore, if we consider an average error, it is reasonable to take half of the error to be the average error, which is 0.075. This result is approximately equal to the measured result, which is 0.09±0.02.

## 5. Discussion

In Figure 14, we observed unbalanced coefficients in the relationship between positive and negative magnetic field regions, and the rapid damping of the spin rate for magnetic fields of −459 nT and −510 nT. We believe that this is due to the strongly coupled Cs electron spin and ${}^{131}$Xe nuclear spins as the resonance frequencies of the two approach each other [43,44].

As shown in Table 1, the measured coil constants are different based on the methods. The Coil method and the Precession method can obtain similar coil constants results. However, the theoretical results have obvious differences from the other two methods. We believe that the differences are from several factors. The design of the coil assumes neglected thickness of the current plane. However, the coils are fabricated by flexible printed circuit. Three sets of bi-planar coils are seated on each other. There is noticeable thickness of the coils. Thus, the thickness of the coils could lead to the deviation. Moreover, the coils are mounted on the 3D-printed PEEK sensor head. The machining tolerance of the sensor head could lead to distances between the coils deviated from the theoretical design. This could lead to the measured coil constants’ deviation from the theoretical results. In the references, machining tolerances should be considered to obtain a very uniform magnetic field [45].

The bi-planar coil in the *z* direction generates a magnetic field perpendicular to the pumping laser direction. As a result, the single-beam configured atomic magnetometer is only sensitive to magnetic fields perpendicular to the laser direction, depending on the axis of modulation. If the holding magnetic field is in the *z* direction, there will be a large background magnetic field that will degrade the magnetometer’s sensitivity. Therefore, it is almost impossible to measure the magnetic field produced by the nuclear spin rotating in the xy plane. To measure the precession nuclear spin signal in the *x* direction, we chose the sensitive axis to be in the *x* direction, and according to theory, we will obtain a precession nuclear spin signal in the *x* direction when we suddenly change the holding magnetic field from the pumping laser *y* direction to the *z* direction. However, due to the degradation of the magnetometer’s sensitivity resulting from changing the holding magnetic field produced by the *z* direction bi-planar coil, it was difficult to measure the precession relaxation time of the nuclear spin, and we could not measure the magnetic field gradient produced by the *z* bi-planar coil. Our results show that the calculated average magnetic field gradient of the *x* or *y* direction bi-planar coil is 0.08, and our measured magnetic field gradient by the in situ method is 0.09±0.02. The measured result coincides with the theoretical result.

## 6. Conclusions

In summary, we successfully developed a miniature bi-planar coil system that is suitable for chip-scale atomic devices applications. Due to its small size and the ability to produce a three-dimensional magnetic field by confinement of the current in only two planes, the bi-planar coil could greatly reduce the size of atomic devices if integrated. However, when miniaturized, it is important to consider the magnetic field gradients produced by the bi-planar coil, and measuring them is crucial for improving the design. The traditional method for measuring the magnetic field gradients in such a small area is difficult. Therefore, we developed an in situ method based on the relaxation of ${}^{131}$Xe nuclear spins, which proved to be effective. Our results show that the designed magnetic field gradient for the *x* or *y* bi-planar coil is 8%, and our measured magnetic field gradient is 9%. We also measured the coil constants using both calibration and nuclear spin precession methods.

## Figures and Tables

**Figure 1 micromachines-14-01985-f001:**
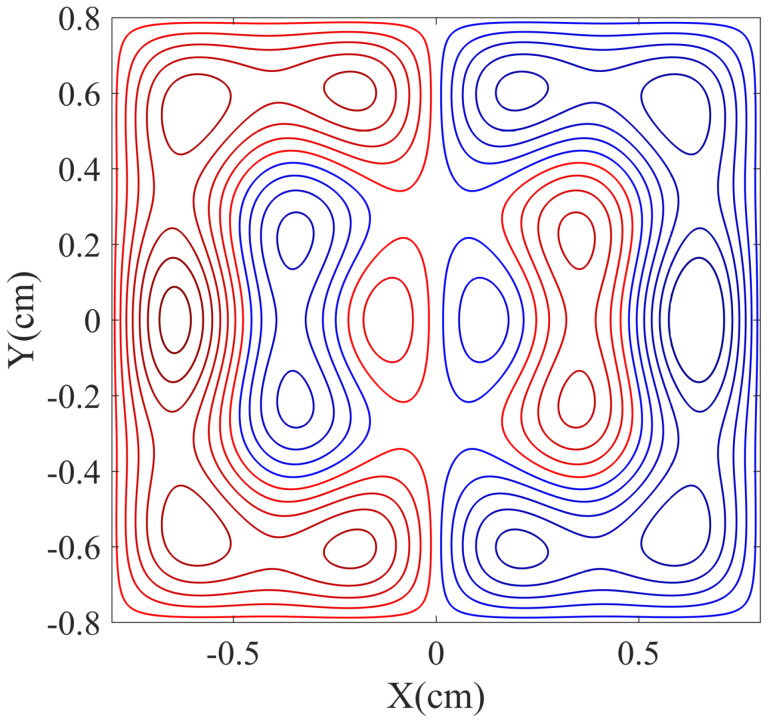
The figure shows the design of the bi-planar coil for the *x* and *y* directions, where the red and blue lines represent different directions of current flow.

**Figure 2 micromachines-14-01985-f002:**
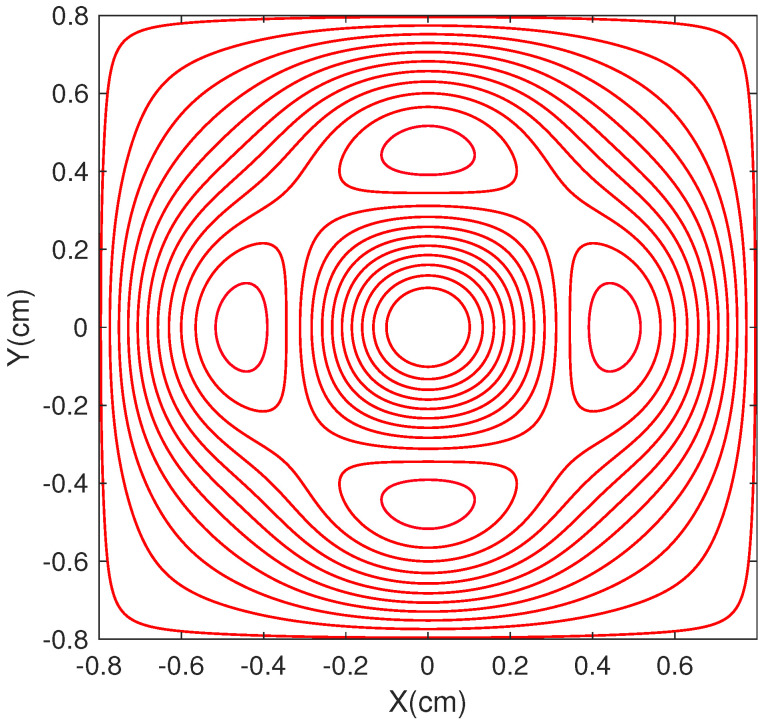
The figure displays the design of the bi-planar coil for the *z* direction, where the red lines represent the same direction of current flow.

**Figure 3 micromachines-14-01985-f003:**
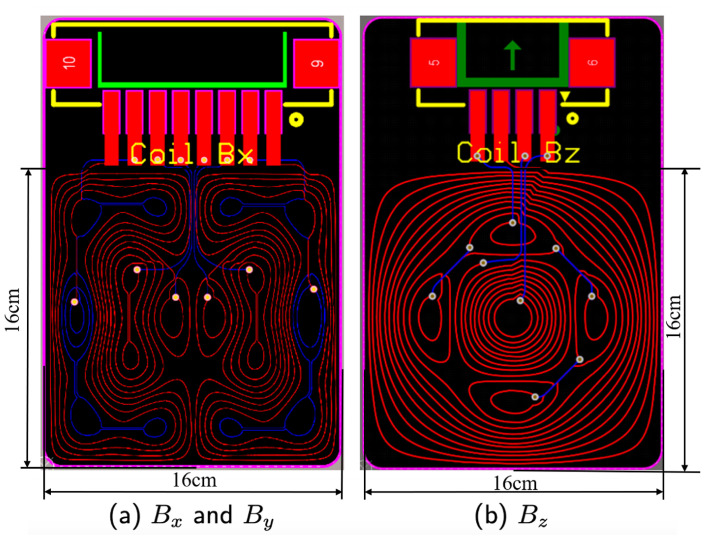
The PCB drawing of $B_{x}$, $B_{y}$ and $B_{z}$

**Figure 4 micromachines-14-01985-f004:**
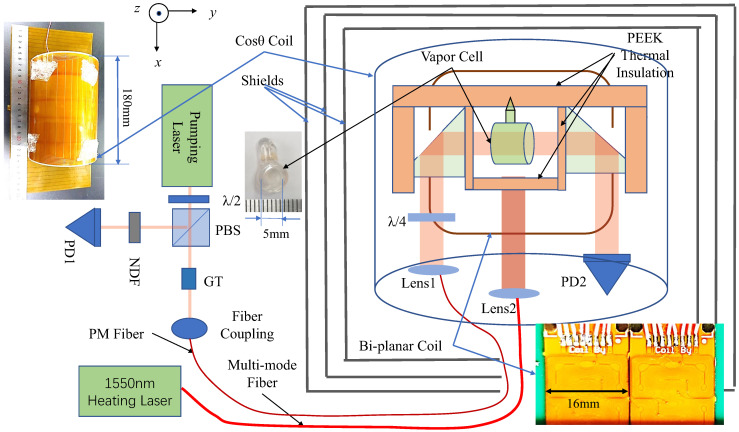
The experimental setup for testing the bi-planar coil. PD: photo diode, NDF: neutral density filter, PBS: polarization beam splitter, GT: Glan–Taylor prism, PEEK: polyether ether ketone, PM: polarization maintaining. The bi-planar coils are set in the xy plane. The two planes of the bi-planar coils are in parallel. The three pair bi-planar coils can produce three dimension magnetic fields. The cos(θ) coils are in a cylinder shape. The cylinder axial direction is the same as the heating laser beam direction. The cos(θ) coils [39] are arranged in such a way that they can produce *z* and *y* direction homogeneous magnetic fields. The cylinder also configures a Lee-Whiting coil, which could produce an *x* direction homogeneous magnetic field [40].

**Figure 5 micromachines-14-01985-f005:**
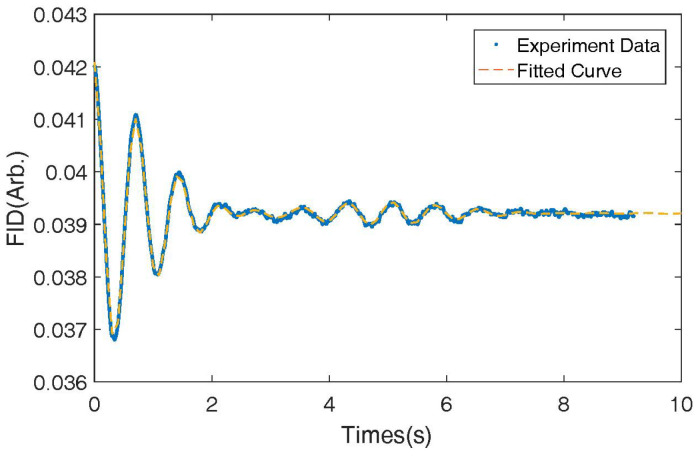
Typical Free Induction Decay (FID) signal of ${}^{131}$Xe nuclear spins under a holding magnetic field produced by the bi-planar coil $B_{y}$.

**Figure 6 micromachines-14-01985-f006:**
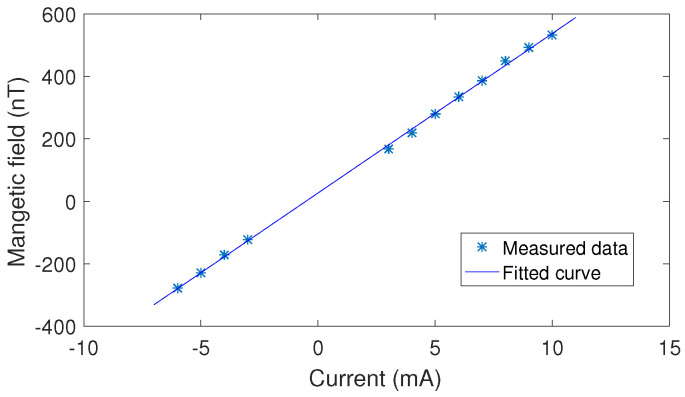
The relationship between the current in the bi-planar coil and the magnetic field strength measured by the precession frequency of ${}^{131}$Xe nuclear spins for the *x* or *y* direction.

**Figure 7 micromachines-14-01985-f007:**
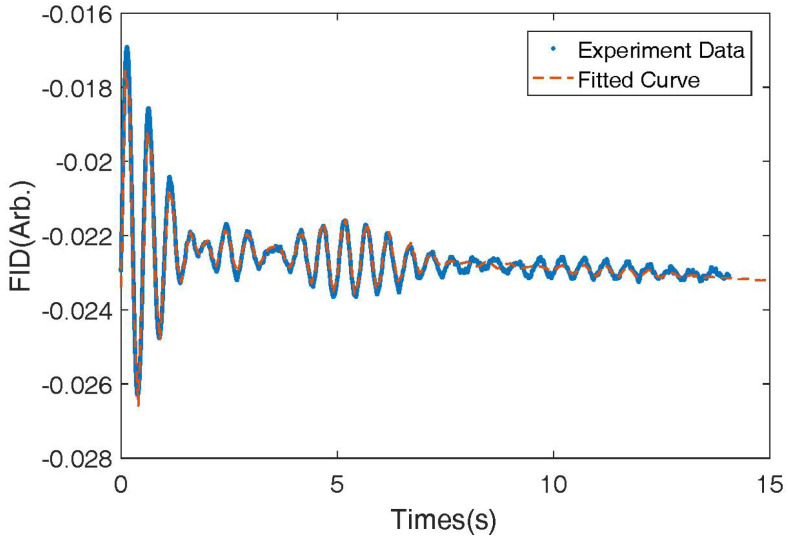
The FID signal of the ${}^{131}$Xe nuclear spins under a *y* direction bigger coil (cos(θ) coil) system with more uniform magnetic field gradients. The nuclear quadrupolar interaction induced beating signals can be resolved. The relaxation of the nuclear spins is smaller.

**Figure 8 micromachines-14-01985-f008:**
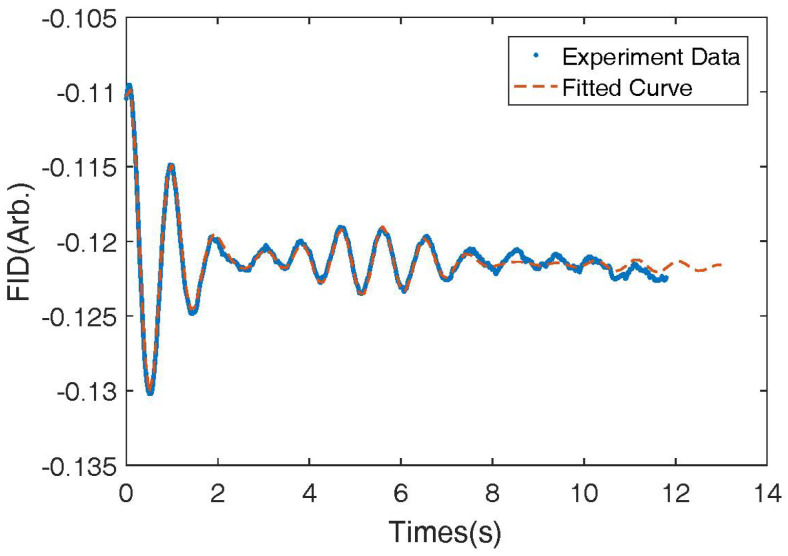
The FID signal of the ${}^{131}$Xe nuclear spins under a bigger coil system with more uniform magnetic field gradients. The magnetic field was in the *z* direction. The relaxation rate was 0.30 $s^{-1}$.

**Figure 9 micromachines-14-01985-f009:**
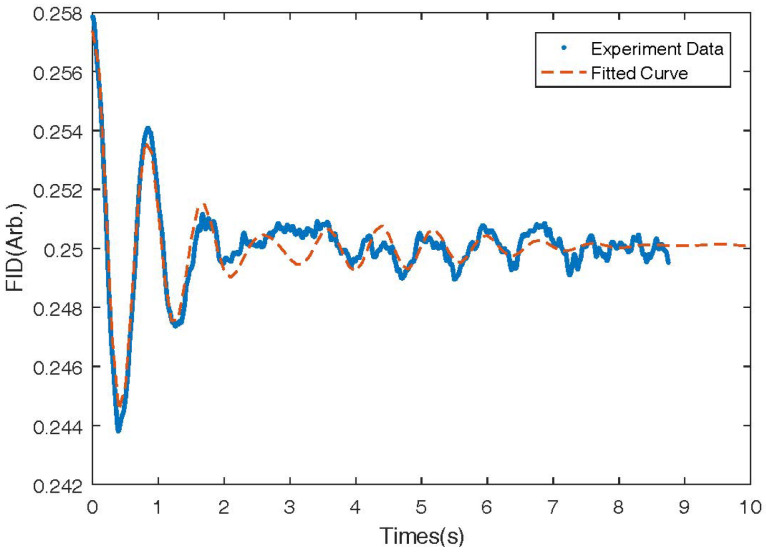
The FID signal of the ${}^{131}$Xe nuclear spins under the *z* bi-planar coil. The relaxation rate was 0.50 $s^{-1}$.

**Figure 10 micromachines-14-01985-f010:**
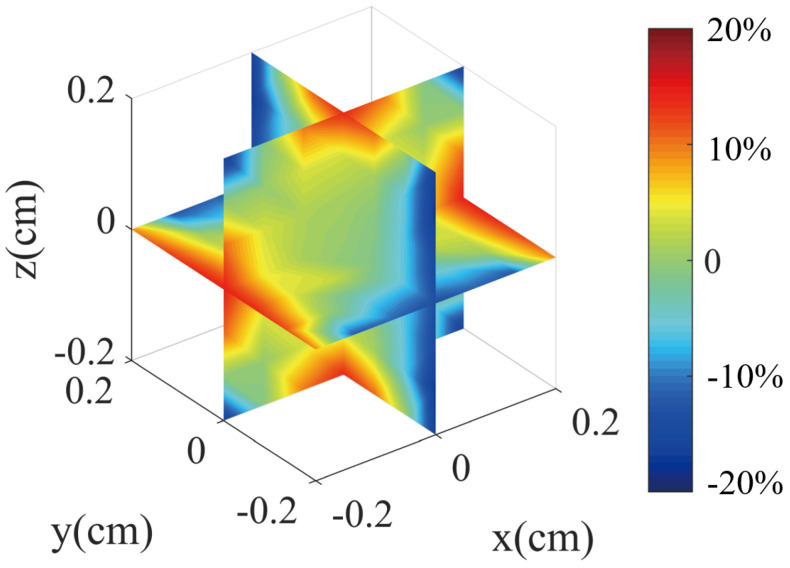
The magnetic field gradient produced by the *y* direction bi-planar coils. Due to symmetry, the *x* direction field gradient is similar. The color represents the error of the magnetic field, which is defined in Equation (Equation 2).

**Figure 11 micromachines-14-01985-f011:**
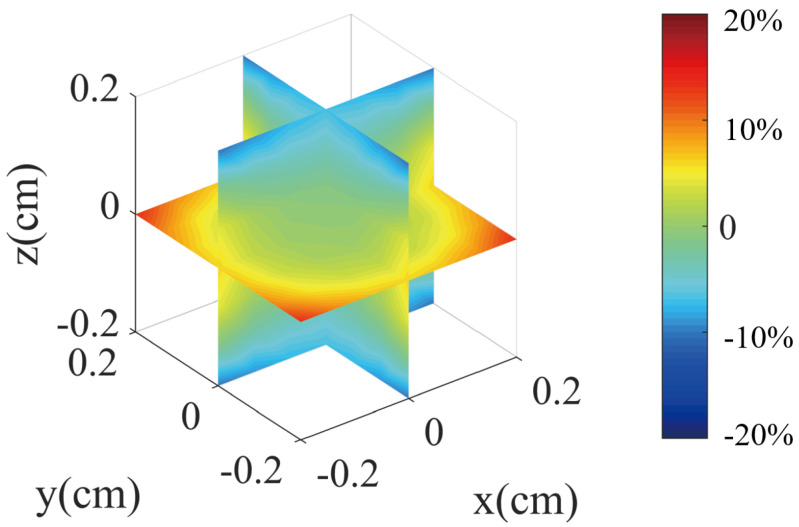
The magnetic field gradient produced by the *z* direction bi-planar coil.

**Figure 12 micromachines-14-01985-f012:**
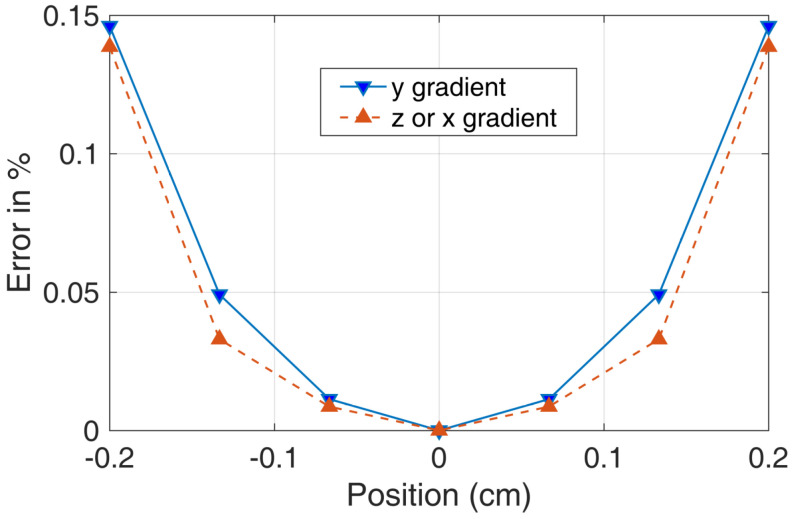
The more detailed plotting of the magnetic field gradient produced by the *y* direction bi-planar coil. ‘y gradient’ represents $\partial B_{y}/\partial y$ and ‘z or x gradient’ represents ‘$\partial B_{z}/\partial z$ or $\partial B_{x}/\partial x$ in the figure. Due to symmetry, the *x* direction bi-planar coil is similar.

**Figure 13 micromachines-14-01985-f013:**
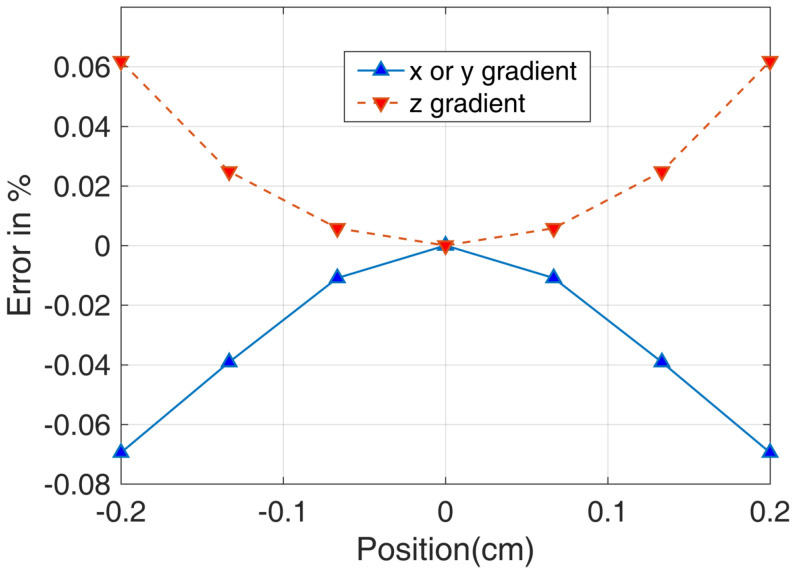
The more detailed plotting of the magnetic field gradient produced by the *z* direction bi-planar coil. ‘z gradient’ represents $\partial B_{z}/\partial z$ and ‘x or y gradient’ represents $\partial B_{x}/\partial x$ or $\partial B_{y}/\partial y$ in the figure.

**Figure 14 micromachines-14-01985-f014:**
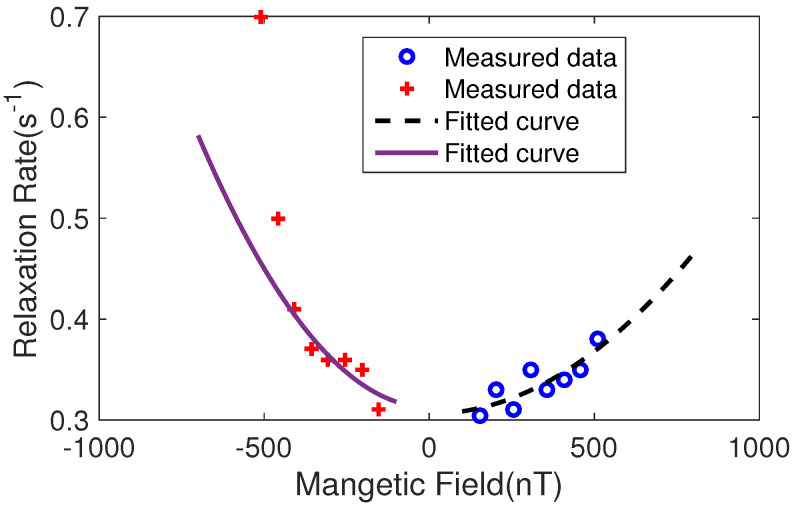
The relation between the magnetic field and the relaxation rate for the *y* bi-planar coil. Here, we did not directly measure the relationship between the magnetic field gradient and the relaxation rates. However, we directly measured the relationship between the magnetic field and relaxation rate because the magnetic field gradient can be calculated to be $\epsilon_{avg}B/0.2$ cm in our design. The *x* bi-planar coil is similar.

**Table 1 micromachines-14-01985-t001:** The measured coil constants of the bi-planar coils by different methods.

Direction	Coil ${}^{a}$	Precession ${}^{b}$	Theoretical
$B_{y}/B_{x}$	49 nT/mA	51 nT/mA	42 nT/mA
$B_{z}$	535 nT/mA	574 nT/mA	476 nT/mA

${}^{a}$ This method is based on using a calibrated coil which produces a standard magnetic field. The responses of the Cs magnetometer to magnetic fields of the calibration coil and the bi-planar coil are compared. ${}^{b}$ This method is based on the precession of ${}^{131}$Xe nuclear spin under several magnetic fields produced by the bi-planar coil.

**Table 2 micromachines-14-01985-t002:** The measured coil constants of the bi-planar coil by calibration coil method.

Item	$B_{x}$ (Lee-Whiting)	$B_{x}$ (bi)	$B_{z}$ (cos)	$B_{z}$ (bi)
Slope of curve	5979 mV/mA	1226 mV/mA	2479 mV/mA	16,600 mV/mA
Coil constant	240 nT/mA	49 nT/mA	80 nT/mA	535 nT/mA

**Table 3 micromachines-14-01985-t003:** Relaxation rate of ${}^{131}$Xe under the big coil and the bi-planar coil.

Coil Type ${}^{a}$	$B_{y}$ and $B_{x}$ Relaxation	$B_{z}$ Relaxation
Big coil	0.30 $s^{-1}$	0.30 $s^{-1}$
Bi-planar coil	0.50 $s^{-1}$	0.50 $s^{-1}$

${}^{a}$ There are two types of coils utilized. The first one is a big coil, which has better uniformity. The other is the bi-planar coil, which is small and need to be evaluated.

## Data Availability

Data underlying the results presented in this paper are not publicly available at this time but may be obtained from the authors upon reasonable request.

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
