# Peer review of "In Situ Study of the Magnetic Field Gradient Produced by a Miniature Bi-Planar Coil for Chip-Scale Atomic Devices"

_micromachines, 2023, doi:10.3390/mi14111985_

Round 1
Reviewer 1 Report
The manuscript should be reduced because there are many repetitions in it.
1. Figs. 1 and 2 are represented at Fig. 4 in a more informative form.
2. The parameters of the vapor сell and some other elements of the experimental setup are repeated many times: on pages 3, 5, 6, 7
3. Eq. (6) contains undefined symbols.
4. Definition of the error in Eq. (2) is incorrect because it defines a deviation.
5. Figures 6 – 8 contain very little information and can be replaced by the presentation of the results in the text.
6. There are many typos.
The authors claim on page 12 that the coils only create gradients ∂Bx/∂x, ∂By/∂y and ∂Bz/∂z, but from the Maxwell equations we have divB=∂Bx/∂x + ∂By/∂y + ∂Bz/∂z =0. Unfortunately, the authors don’t mansion about it. In this case, the definition of these derivatives as magnetic field gradients is incorrect. This fact should be taken into account when discussing the relaxation on page 13.
Reviewer 2 Report
Please address next comments:
- Include a 3D or 2D image of the coils design. Figures 1 and 2 claims to show the design but it is not totally clear how the design is done. Fig 1 and 2 seems more a simulation than a design, which are the current tracks line, what lines are conductive and what are insulating? Provide a more clear description of the part to be machined. Where does the current get into the coil and where does it exit. Which are those connections points.
- Could you provide mor information about the calculations on the magnetic field shown in fig 12 and 13? It is purely analyticial, FEM? Could you add the coils design it self? provide cross sections
Reviewer 3 Report
The manuscript describes an approach to measure the magnetic field gradient of a bi-planar miniature coil. Results obtained in the paper may be used in designing various atomic devices, including magnetic resonance imaging systems. The manuscript makes positive impression, it is well written and readable. Authors provide analytical calculations, finite element simulation and experimental results on the magnetic field of the coil, which support the conclusions. The paper is interesting for large area of science, including biology and medicine. In my opinion, it can be published in Micromachines after minor corrections:
1. In the introduction, I recommend highlighting the scientific novelty a bit stronger. Is the proposed method for masuring the magnetic field gradient novel? If not, what new ideas do the authors propose?
2. In page 6, the authors say “make it more conducive to miniaturization”. I recommend saying “make it more suitable for miniaturization”.
Round 2
Reviewer 1 Report
The authors tok into account allmy wishes and answered the comments in a reasoned manner. The article can be accepted for publication.